# Loss of PARP7 Increases Type I Interferon Signaling in EO771 Breast Cancer Cells and Prevents Mammary Tumor Growth by Increasing Antitumor Immunity

**DOI:** 10.3390/cancers15143689

**Published:** 2023-07-20

**Authors:** Marit Rasmussen, Karoline Alvik, Vinicius Kannen, Ninni E. Olafsen, Linnea A. M. Erlingsson, Giulia Grimaldi, Akinori Takaoka, Denis M. Grant, Jason Matthews

**Affiliations:** 1Department of Nutrition, Institute of Basic Medical Sciences, Faculty of Medicine, University of Oslo, Sognsvannsveien 9, 0372 Oslo, Norway; marit.rasmussen@medisin.uio.no (M.R.); karoline.alvik@medisin.uio.no (K.A.); n.e.olafsen@medisin.uio.no (N.E.O.); l.a.m.erlingsson@medisin.uio.no (L.A.M.E.); 2Department of Pharmacology and Toxicology, University of Toronto, 1 King’s College Circle, Toronto, ON M5S 1A8, Canada; vinicius.kannen@utoronto.ca (V.K.); denis.grant@utoronto.ca (D.M.G.); 3Faculty of Life Sciences, University of Bradford, Bradford BD7 1DP, UK; g.grimaldi@bradford.ac.uk; 4Division of Signaling in Cancer and Immunology, Institute for Genetic Medicine, Hokkaido University, 7 Chome Kita 15 Jonishi, Sapporo 060-8628, Japan; takaoka@igm.hokudai.ac.jp

**Keywords:** PARP7, ADP-ribosylation, type I interferon, tumor immunity, breast cancer

## Abstract

**Simple Summary:**

Cancer development depends on interactions between the tumor microenvironment and the immune system. PARP7 negatively regulates the type I interferon pathway, ultimately preventing immune cells from detecting and eliminating cancer cells. Recently, inhibition of PARP7 activity has been shown to restore type I interferon signaling, resulting in tumor regression. Here we investigated the effect of stable PARP7 knockout in mammary cancer cells and used a genetic mouse model to study the effects of PARP7 loss on tumor growth in vivo.

**Abstract:**

PARP7 is a member of the ADP-ribosyltransferase diphtheria toxin-like (ARTD) family and acts as a repressor of type I interferon (IFN) signaling. PARP7 inhibition causes tumor regression by enhancing antitumor immunity, which is dependent on the stimulator of interferon genes (STING) pathway, TANK-binding kinase 1 (TBK1) activity, and cytotoxic CD8^+^ T cells. To better understand PARP7′s role in cancer, we generated and characterized PARP7 knockout (Parp7^KO^) EO771 mouse mammary cancer cells in vitro and in a preclinical syngeneic tumor model using catalytic mutant *Parp7^H532A^* mice. Loss of PARP7 expression or inhibition of its activity increased type I IFN signaling, as well as the levels of interferon-stimulated gene factor 3 (ISGF3) and specifically unphosphorylated-ISGF3 regulated target genes. This was partly because PARP7′s modification of the RelA subunit of nuclear factor κ-B (NF-κB). PARP7 loss had no effect on tumor growth in immunodeficient mice. In contrast, injection of wildtype cells into *Parp7^H532A^* mice resulted in smaller tumors compared with cells injected into *Parp7^+/+^* mice. *Parp7^H532A^* mice injected with Parp7^KO^ cells failed to develop tumors and those that developed regressed. Our data highlight the importance of PARP7 in the immune cells and further support targeting PARP7 for anticancer therapy.

## 1. Introduction

Enhancing immune responses by increasing proinflammatory cytokine levels or by inhibiting immune suppressive checkpoint signaling has led to unprecedented clinical outcomes for many cancer patients [1]. However, not all patients respond, and for many tumor types immune cell infiltration is poor, rendering immunotherapeutic strategies ineffective. Thus, new strategies are needed to increase tumor inflammation and immune cell infiltration to improve cancer immunotherapy.

Many cancer cells exhibit an abnormal number of nucleic acids in the cytoplasm, which can activate pattern recognition receptors (PRRs) and pathways, such as the cyclic GMP-AMP synthase (cGAS)-stimulator of interferon genes (STING) pathway [2]. Upon binding of cytoplasmic DNA, cGAS catalyzes the synthesis of 2′3′-cyclic GMP-AMP (cGAMP), activating STING. STING activates TANK-binding kinase 1 (TBK1), which phosphorylates and activates interferon (IFN) regulatory factor 3 (IRF3) and nuclear factor κ-B (NF-κB), increasing the expression of type I IFNs, such as IFN-β and proinflammatory cytokines and chemokines [2]. Secreted IFN-β binds to the heterodimeric IFN-α and -β receptors (IFNAR1 and 2), resulting in the activation of signal transducers and activators of transcription 1 (STAT1), STAT2, and IRF9, which together form the trimeric IFN stimulated gene factor 3 (ISGF3) complex. ISGF3 induces the expression of many inflammatory cytokines, including C-X-C motif chemokine ligand 10 (CXCL10), which modulates host responses to tumors. The cGAS-STING pathway and the stimulation of type I IFNs are critical for antitumor immune responses [3,4,5,6]. Because of this, numerous natural and synthetic STING agonists have been studied in different preclinical models and clinical settings. 5,6-Dimethylxanthenone-4-acetic acid (DMXAA) was the first to show promising antitumor activity in murine models, but it failed in clinical trials because it activates murine and not human STING [7]. These results prompted the development of more stable and potent compounds targeting human STING, such as ADU-S100/MIW815, which is in clinical trials for patients with advanced/metastatic cancers [8].

The ADP-ribosyltransferase diphtheria toxin-like (ARTD) family consists of 17 members that use β-nicotinamide adenine dinucleotide (NAD^+^) as a substrate to catalyze the transfer of ADP-ribose onto target proteins [9,10]. Most ARTD family members, known as mono-ADP-ribosyltransferases (mono-ARTs), catalyze the transfer of a single unit of ADP-ribose onto their substrates, referred to as MARylation. Other members, known as poly-ARTs, attach polymers of ADP-ribose units onto their substrates in a process known as PARylation [9]. PARP1 is a poly-ART, and PARP1 inhibition is an effective anticancer strategy that takes advantage of cellular stress induced by DNA damage and defective DNA repair pathways [11]. The potential for other PARPs to become viable cancer therapeutic targets has gained significant interest in recent years, and several inhibitors of other ARTD family members have been developed [10,12].

PARP7, also known as 2,3,7,8-tetrachlorodibenzo-*p*-dioxin poly-ADP-ribose polymerase (TIPARP), is a mono-ART that plays a role in many cellular pathways, including stem cell pluripotency, viral replication, neuronal function, and gene regulation [13,14,15]. PARP7 is induced by cellular stresses and by ligand activated aryl hydrocarbon receptor (AHR) and, in turn, acts as a negative regulator of AHR signaling [16]. Multiple lines of evidence also point to PARP7 as a critical regulator of innate and adaptive immune signaling [17]. PARP7 inhibits type I IFN signaling by MARylating and inactivating TBK1 [15], which in cancer acts to repress antitumor immunity, enabling cancer cells to evade immunosurveillance and proliferate. This has led to the development of several selective PARP7 inhibitors, including KMR-206 [18,19], RBN-2397 [12], and I-1 [20]. All PARP7 inhibitors enhance type I IFN signaling in vitro, and for certain cell types, such as NCI-H1373, they are also antiproliferative [12]. CRISPR/Cas9 screening revealed that intact innate immune signaling and AHR expression are necessary for the antiproliferative effects of PARP7 inhibition in vitro [21]. CT26 colon carcinoma cells are resistant to the antiproliferative effect of PARP7 inhibition, but treatment with RBN-2397 or I-1 reduces tumor growth in a syngeneic mouse model. PARP7 inhibitor-mediated tumor regression is dependent on an intact type I IFN signaling pathway and increased infiltration of CD8^+^ T cells [12,20]. RBN-2397 is currently the only PARP7 inhibitor in clinical trials as a monotherapy in patients with advanced solid tumors (NCT04053673) and as combination treatment with the immune checkpoint inhibitor pembrolizumab (NCT05127590).

Despite convincing evidence for the antitumor action of PARP7 inhibition in many preclinical models, the impact of PARP7 inhibition in breast cancer is not fully determined. We previously reported that PARP7 negatively regulates estrogen receptor α signaling in vitro [22], and other studies found that PARP7 knockdown promoted tumor growth in an MCF-7 xenograft model [23]. However, these studies were performed in the absence of immune cells or a functional immune system. Whether the reported antitumor effects of PARP7 inhibition is due to the inhibition of PARP7 activity in tumor or immune cells, or both, is not fully resolved. In the current study, we generated and characterized PARP7 knockout (Parp7^KO^) EO771 mouse mammary cancer cells in vitro and in a preclinical syngeneic tumor model using catalytical mutant *Parp7^H532A^* mice. Loss of PARP7 function in either cancer cells or recipient mice resulted in reduced tumor growth, with the loss of PARP7 activity in the recipient having a greater reduction in tumor growth. Our findings further support targeting PARP7 inhibition for cancer therapy.

## 2. Materials and Methods

### 2.1. Chemicals and Ligands

Dimethyl sulfoxide (DMSO) and β-mercaptoethanol were purchased from Sigma-Aldrich (St. Louis, MO, USA). The murine STING ligand, DMXAA, and the TBK1/IKKε inhibitors, MRT67307 and BX795, were purchased from Invivogen (San Diego, CA, USA).

### 2.2. Plasmids

The pSpCas9(BB)-2A-Puro (PX459) (plasmid #62988), p50 cFlag pcDNA3 (plasmid #20018), and RelA cFlag pcDNA3 (plasmid #20012) were obtained from Addgene (Watertown, MA, USA). To create the pEGFP-PARP7 plasmid, mouse PARP7 was amplified with PCR using pCMV6-kan/neo-mTiparp as a template (Origene, Rockville, MD, USA) and the forward primer 5′-CAAAGAATTCATGGAAGTGGAAACCACTGAACC-3′ and reverse 5′-CAAAGTCGACTCAAATGGAAACAGTGTTACTGACT-3′. Restriction enzyme sites used for cloning are underlined in the primer sequences. Following amplification, the PARP7 sequence was cloned into EcoRI and SalI sites of pEGFP-C2 (Clontech, Mountain View, CA, USA). pEGFP-PARP7H532A was created by PCR mutagenesis using forward primer 5′-GAGAGACACTTATTT**GC**TGGAACATCCCAAGA-3′ and reverse 5′-TCTTGGGATGTTCCA**GC**AAATAAGTGTCTCTC-3/. The introduced mutation is shown in bold font.

### 2.3. Cell Culturing

EO771 and NCI-H1373 cells (ATCC) were maintained in RPMI (1.0 g/L glucose), supplemented with 10% *v*/*v* heat-inactivated fetal bovine serum (FBS), 1% *v*/*v* L-glutamine, and 1% *v*/*v* penicillin–streptomycin (P/S). COS-1 cells (ATCC) and MEFs (described elsewhere [16,24]) were maintained in DMEM (1.0 g/L glucose) supplemented with 10% *v*/*v* FBS, 1% *v*/*v* L-glutamine, and 1% *v*/*v* P/S. Cells were cultured at 37 °C with 100% humidity and 5% CO_2_. When confluency reached 80%, cells were subcultured. Cells were seeded at a density of 1 × 10^5^ cells per mL cell media 24 h prior to experiments unless otherwise stated.

### 2.4. Western Blotting

Cells were seeded in six-well plates. The following day, cells were treated with test ligands. The cells were washed with PBS, scraped, and pelleted before lysis with 1× RIPA buffer supplemented with fresh 1× PIC. For experiments involving detection of phosphorylated proteins, the cells were washed with PBS prior to lysis in 95 °C TE-buffer supplemented with 1% *w*/*v* SDS. Samples were sonicated at a high intensity for 2 × 30 s on/off and boiled at 95 °C for 10 min. After the lysates were clarified, the protein concentration was determined with BCA assays (Thermo Fisher Scientific, Waltham, MA, USA). Proteins were separated using SDS-PAGE and transferred to polyvinylidene fluoride (PVDF) membranes. The nuclear and cytoplasmic fractions were prepared using the NE-PER™ Nuclear and Cytoplasmic Reagents kit (Thermo Fisher Scientific) according to the manufacturer’s instructions. The antibodies used were anti-AHR (Enzo Life Sciences, Farmingdale, NY, USA; bml-sa210-0100), anti-cGAS (Invitrogen, Waltham, MA, USA; 10H1L5), anti-pSTING (S365) (Cell Signaling Technology, Danvers, MA, USA; D8F4W), anti-STING (Cell Signaling Technology; D2P2F), anti-pTBK1 (S172) (Cell Signaling Technology; D52C2), anti-TBK1 (Cell Signaling Technology; E9H5S), anti-pIRF3 (S396) (Cell Signaling Technology; D601M), anti-IRF3 (Cell Signaling Technology; D83B9), anti-pSTAT1 (Y701) (Cell Signaling Technology; D4A7), anti-STAT1 (Cell Signaling Technology; #9172), anti-STAT2 (Cell Signaling Technology; D9J7L), anti-IRF9 (Cell Signaling Technology; D9I5H), anti-β-actin (Sigma-Aldrich; AC-74), anti-Lamin A/C (Cell Signaling Technology; #2032), anti-α-tubulin (Sigma-Aldrich; T5168), anti-NF-kappaB p105/p50 (Cell Signaling Technology; D4P4D), and anti-NF-kappaB p65 (Cell Signaling Technology; D14E12). To detect PARP7, we used our lab generated mouse monoclonal anti-PARP7 antibody that has been previously described [22]. Briefly, recombinant 6x histidine-tagged murine PARP7 protein (amino acids 1-320) was injected into eight-week-old female BALB/c mice at 2-week intervals. Hybridomas were generated, and those producing specific antibodies against murine PARP7 were selected by ELISA.

After incubation with secondary antibody (Rabbit, Mouse, Cell Signaling Technology), bands were visualized with SuperSignal™ West Dura Extended Duration Substrate or SuperSignal™ West Atto Ultimate Sensitivity Substrate (Thermo Fisher Scientific).

### 2.5. Real Time qPCR (RT-qPCR)

RNA was isolated using the Aurum™ Total RNA isolation kit (BioRad, Hercules, CA, USA) and used as a template to synthesize cDNA with the High-Capacity cDNA Reverse Transcription Kit (Applied Biosystems, Waltham, MA, USA). The RT-qPCR was set up as previously described [22]. The primers used are provided in Appendix A.

### 2.6. Proliferation Assays

Cells were seeded in 96-well plates on day 0 at a density of 2000 cells per well. On day 1, cells were dosed accordingly and placed in the IncuCyte instrument (Sartorius, Göttingen, Germany). The cell proliferation was measured as an increase in cell confluency over time until 100% confluency was reached.

### 2.7. Generation of EO771 Parp7 Knockout Cells

EO771 Parp7^KO^ cells were generated using CRISPR/Cas9. The guide oligos were 5′-CACCGTCTTCTCAGAAATTCTCATT-3′ and 5′-AAACAATGAGAATTTCTGAGAAGAC-3′. The resulting gRNA was cloned into the PX459 plasmid, and transfection, selection, and T7 assays were carried out essentially, as we previously described [22]. Genomic DNA from multiple clones was harvested, and the region surrounding the target site of the *Parp7* (also known as *Tiparp*) gene was amplified and sequenced to confirm knockout. The primers used were: Forward 5′-TGCAGATTTTTGCATAGCTTTTG-3′ and reverse 5′-TTGTCTTGGAAAGCTCCTGGT-3′. After screening, multiple clones were chosen and subsequently expanded and further analyzed. EO771 cells were also transfected with empty PX459 plasmid and treated with puromycin before being expanded. However, unlike the CRISPR knockout clones, pooled cell populations rather than individual cell clones were used. The EO771 PX459 cells are referred to as EO771 WT cells in this study.

### 2.8. Spheroid Formation

Cells were plated in 96-well Clear Round Bottom Ultra-Low Attachment Microplate (Corning, Corning, NY, USA) at a density of 500 cells per well and pelleted at the bottom of the well. The following day, the cell culture medium was replaced with a solution of 2% *v*/*v* Geltrex™ LDEV-Free Reduced Growth Factor Casement Membrane Matrix (Gibco, Waltham, MA, USA) diluted in cell culture media. After 7 days of culturing, with regular replenishments of cell culture media, spheroid formation was monitored and measured with a ZEISS Axio Vert.A1 inverted microscope with Axiocam 305 mono camera (ZEISS, Oberkochen, Germany).

### 2.9. ELISA

The cell culture media from treated cells were collected and stored at −20 °C until analysis. The ELISA was carried out using the DuoSet^®^ Mouse IFN-β kit (R&D Systems, Minneapolis, MN, USA) according to the manufacturer’s instructions.

### 2.10. Co-Immunoprecipitation

Cos-1 cells were transfected with 0.5 µg of p50 cFlag pcDNA3 or RelA cFlag pcDNA3 alone or along with either 1 µg of pEGFP-PARP7 or 0.8 µg of pEGFP-PARP7H532A using Lipofectamine™ 2000 (Invitrogen). After 24 h, cell lysis and co-immunoprecipitation was carried out as previously described [22]. The antibodies used for detection with Western blotting were anti-poly/mono-ADP-ribose (Cell Signaling Technology; E6F6A), anti-GFP (rabbit) (Abcam, Cambridge, United Kingdom; ab290), anti-GFP (mouse) (Clontech Laboratories, Mountain View, CA, USA; JL-8), anti-FLAG (rabbit) (Sigma Aldrich; F7425), and anti-FLAG (mouse) (Sigma-Aldrich; M2).

### 2.11. RNAi Knockdown

Gene silencing was performed by transfecting EO771 cells with 25 nM of ON-TARGETplus Non-targeting Control Pool (Dharmacon, Lafayette, CO, USA; #D-001810-10-20), siRNA Tbk1 Mouse SMARTPool (Dharmacon; #L-063162-00-0010), siRNA Rela Mouse SMARTPool (Dharmacon; #L-040776-00-0010), or combinations of these, using Lipofectamine™ RNAiMAX (Invitrogen) according to the manufacturer’s instructions. After 48 h, cell lysates were prepared and subject to SDS-PAGE and Western blotting to monitor gene silencing. The remaining cells were treated for 2 h with appropriate ligands. RNA isolation, cDNA synthesis, and RT-qPCR were performed as described previously [22].

### 2.12. Mouse Models and Tumor Studies

Immunodeficient NOD.Cg-*Prkdc^scid^ Il2rg^tm1Wjl^/SzJ* mice (Strain #005557), commonly referred to simply as NSG-mice, were purchased from The Jackson Laboratory (Farmington, CT, USA). Female 8-week old NSG-mice were subjected to a single subcutaneous injection in the fourth mammary fat pad on the right side with either EO771 WT or Parp7^KO^ cells, *n* = 8. On the day of the cell injection, the cells were trypsinized and resuspended in 0.9% sterile saline solution, at a concentration of 3.33 × 10^6^ cells/mL. The injection volume was 150 µL, resulting in 5 × 10^5^ cells per injection. The mice were put under isoflurane anesthesia at the time of injection and observed for any ill effects in the time following. At day 8, tumors were measured with calipers under anesthesia, and the tumor volume was estimated using the standard formula: π/6 × W^2^ × L. Tumors were measured daily until they reached a volume of 400 mm^3^ at which point the mice were euthanized by cervical dislocation.

The generation of the *Parp7^H532A^* mouse strain has been described previously [24]. Female 8-week old *Parp7^+/+^* and *Parp7^H532A^* mice were subjected to a single subcutaneous injection in the fourth mammary fat pad on the right side of either EO771 WT cells or Parp7^KO^ cells. This resulted in 4 different experimental conditions, with *n* = 6–12: *Parp7^+/+^* mice with EO771 WT cells, *Parp7^+/+^* mice with EO771 Parp7^KO^ cells, *Parp7^H532A^* mice with EO771 WT cells and, finally *Parp7^H532A^* mice with EO771 Parp7^KO^ cells. At day 7, tumors were measured, and the volume was estimated as described above. The tumors were measured three times per week until they reached a volume of 400 mm^3^ or until day 30 at which point the mice were euthanized by cervical dislocation, and tumor tissues were harvested for genetic and histological analysis. The animals were kept at the Section of Comparative Medicine at the Institute of Basic Medical Sciences, University of Oslo, under a 12 h light/dark cycle, with ad libitum access to standard rodent chow and drinking water. All animal experimental procedures were approved and registered at the Norwegian Food and Safety Authority and the National Animal Research Authority (FOTS ID: 25766).

For gene expression analyses, frozen tumor tissue was homogenized with glass beads in 700 µL of RNA lysis buffer (BioRad) using the Precellys Tissue Homogenizer (Bertin Technologies, Montigny-le-Bretonneux, France). RNA isolation, cDNA synthesis, and RT-qPCR were carried out as described previously [22].

### 2.13. Histology

Standard methods were used for sectioning and staining for CD3 and CD68. Fixated tumor tissues were provided to the HistoCore Facility at the Princess Margaret Cancer Centre (Toronto, ON, Canada), and sample processing, staining, and scanning was carried out at the facility. Quantification analysis was conducted with QuPath v0.4.3 [25].

### 2.14. CD8^+^ T-Cell Isolation and Activation

Spleens from either *Parp7^+/+^* or *Parp7^H532A^* mice were dissected and immediately transferred to a tube containing ice cold PBS. The spleens were homogenized through a 70 µm cell strainer using a syringe plunger. The resulting cell suspension was washed with PBS, and red blood cells were lysed with eBioscience™ RBC Lysis Buffer (Invitrogen) for 5 min at room temperature. Cells were washed with PBS, counted, and subjected to CD8^+^ T-cell isolation with the Dynabeads™ Untouched™ Mouse CD8 Cells isolation kit (Invitrogen). The resulting CD8^+^ T cells were cultured in RPMI supplemented with 10% *v*/*v* FBS, 1% *v*/*v* L-glutamine, *v*/*v* 1% P/S, 100 µM MEM Non-essential Amino Acid Solution (Sigma-Aldrich), 1 mM sodium pyruvate (Gibco), 10 mM HEPES (Gibco), mouse interleukin-2 (Roche, Basel, Switzerland), and 50 µM β-mercaptoethanol. Half of the cells were activated using Dynabeads™ Mouse T-Activator CD3/CD28 beads (Gibco) according to the manufacturer’s instructions. After activation, cells were plated in 24-well plates and dosed with ligands. The following day, cell culture media were harvested for analysis with ELISA, and RNA isolation, cDNA synthesis, and RT-qPCR were carried out as described previously [22].

### 2.15. Isolation and Polarization of Bone Marrow Derived Macrophages

Bone-marrow-derived macrophages (BMDMs) were isolated according to established protocols [26] and seeded in 24-well plates at a density of 1.0 × 10^6^ cells/mL. The BMDMs were cultured in DMEM supplemented with 10% *v*/*v* FBS, 1% *v*/*v* L-glutamine, 1% *v*/*v* P/S, 10 mM HEPES, and 15% *v*/*v* conditioned L929 media. Cells were washed with PBS, and the medium was changed on days 3 and 6 after isolation. On day 7, cells were polarized by adding of 50 ng/mL lipopolysaccharide (LPS) (Sigma-Aldrich) and 20 ng/mL of IFN-γ (Peprotech, Cranbury, NJ, USA) for the M1 phenotype, and 20 ng/mL of Il-4 (Sigma-Aldrich) for the M2 phenotype. RNA isolation, cDNA synthesis, and RT-qPCR were carried out as described previously [22].

### 2.16. Statistical Analysis

All data are represented as the standard error of the mean (S.E.M) of at least three individual experiments and analyzed with GraphPad Prism v8.2 (San Diego, CA, USA). Statistical analyses were conducted using a two-tailed Student’s *t*-test, one- or two-way analysis of variance (ANOVA).

## 3. Results

### 3.1. PARP7 Loss or Inhibition Increases Type I Interferon Signaling in EO771 Cells

EO771 cells, murine breast carcinoma cells isolated from C57BL/6 mice, were used to examine the function of PARP7 in this study [27,28]. To determine the expression levels of PARP7 in EO771 cells, we treated cells with RBN-2397 to inhibit PARP7 activity and stabilize its protein levels, as we have previously described [22]. PARP7 was detected after RBN-2397 treatment, but only a faint band representing PARP7 was observed in solvent treated cells (Figure 1A). This was also confirmed in RBN-2397-sensitive human NCI-H1373 cells (Figure 1B). NCI-H1373 cells express AHR, and AHR-levels increased in response to RBN-2397. Since previous studies reported that AHR is necessary for the antiproliferative effect of PARP7 inhibition [21], we determined the levels of AHR in EO771 cells. No AHR was detected in the EO771 cells (Figure 1C) when compared with extracts from PyMT cells, a mammary cell line derived from spontaneously arising tumors in transgenic mouse mammary tumor virus-polyoma middle tumor-antigen (MMTV-PyMT) C57BL/6 mice and a PyMT Ahr^KO^ clone. The generation and characterization of the PyMT Ahr^KO^ clone will be described elsewhere. We then determined if EO771 cells were sensitive to the antiproliferative effects of RBN-2397. Treatment with RBN-2397 decreased proliferation at its highest concentration (Figure 1D). We observed a rightward shift in the time to 50% confluency from 63 h for untreated cells to 73 h for 1000 nM RBN-2397-treated cells. This resulted in an approximately 25% reduction in cell proliferation for 1000 nM RBN-2397 compared with the untreated cells at 50% confluency. As expected, both RBN-2397 concentrations reduced proliferation in NCI-H1373 cells (Figure 1E), which expressed AHR (Figure 1B). We next treated EO771 cells with DMXAA to characterize their STING-mediated increases in type I IFN genes, *Ifnb1* and *Cxcl10*. Time course studies revealed that the *Ifnb1* levels peaked after 2 h (Figure 1F), whereas the *Cxcl10* levels peaked after 6 h of treatment (Figure 1G). These data show that EO771 cells are relatively resistant to the antiproliferative effects of PARP7 inhibition, and that they harbor an intact STING regulated type I IFN response.

To examine PARP7′s role in type I IFN signaling in EO771 cells, we generated PARP7 knockout (Parp7^KO^) cells using CRISPR/Cas9. The presence of insertions or deletions resulting in reading frame errors and consequently PARP7 knockout were confirmed by DNA sequencing (Appendix A). We selected two clones (termed clone 1 and 2) to expand and characterize. To verify loss of PARP7 protein in the knockout cell lines, WT and Parp7^KO^ cells were treated with RBN-2397 and PARP7 levels detected by Western blotting (Figure 2A). No PARP7 protein was detected in either Parp7^KO^ clone. Cell proliferation assays revealed that both Parp7^KO^ clones displayed a slight, yet significant, decrease in proliferation compared with WT cells (Figure 2B). We did not observe any differences in proliferation upon RBN-2397 treatment in the knockout cells (Appendix A). We next evaluated whether PARP7 loss affected spheroid formation (Figure 2C). Both Parp7^KO^ clones displayed a significant decrease in spheroid volume compared with WT. These data suggest that PARP7 may affect cellular adhesion or cytoskeletal proteins. In support of this, a previous study reported that PARP7 MARylates α-tubulin in ovarian cancer cells, resulting in microtubule instability [29]. Since the two clones were comparable, we continued with Parp7^KO^ clone 2, which herein will be referred to as Parp7^KO^ cells.

Because PARP7 represses type I IFN signaling [12,15], we determined the mRNA levels of *Ifnb1* after treating WT cells with DMXAA, and/or RBN-2397, and Parp7^KO^ cells with DMXAA for 2 h. Both loss and inhibition of PARP7 resulted in significant increases in *Ifnb1* mRNA and protein levels (Figure 2D,E). DMXAA treatment significantly increased *Cxcl10* mRNA levels in both WT and Parp7^KO^ cells with slightly higher levels observed in WT cells. *Cxcl10* mRNA levels were further increased after RBN-2397 co-treatment in WT cells (Figure 2F). Treatment with RBN-2397 did not affect expression levels of *Ifnb1* nor *Cxcl10* in Parp7^KO^ cells (Appendix A).

To determine if the observed changes in *Ifnb1* and *Cxcl10* levels were unique to EO771 cells, we treated MEFs isolated from WT mice or catalytic mutant PARP7 (*Parp7^H532A^*) mice with DMXAA +/− RBN-2397. *Ifnb1* levels were significantly higher in both untreated and DMXAA-treated *Parp7^H532A^* MEFs (Figure 2G). Like that observed in EO771 cells, DMXAA treatment increased *Cxcl10* mRNA levels in both WT and *Parp7^H532A^* MEFs, with slightly higher levels being observed in WT MEFs. *Cxcl10* mRNA levels were also further increased after RBN-2397 co-treatment in WT MEFs (Figure 2H). Longer RBN-2397 treatment (24 h) resulted in similar expression levels of *Ifnb1* compared with those observed in untreated Parp7^KO^ cells but higher levels of *Cxcl10*.

Similar findings were observed in MEFs isolated from WT mice or *Parp7^H532A^* mice. Since programmed death-ligand 1 (PDL1) is upregulated in response to type I IFNs [4], and because of the recent clinical trial aimed at determining the potential benefit of RBN-2397 in combination with checkpoint inhibition, we determined whether PARP7 affected the expression levels of *Pdl1*. We found that Parp7^KO^ cells displayed increased *Pdl1* mRNA levels in both treated and untreated samples (Figure 2I). IFN-β is known to act in an autocrine manner to suppress proliferation in cancer cells by inducing a prolongation of all phases in the cell cycle [4]. We therefore investigated whether proliferation was affected by DMXAA combined with PARP7 loss. Treatment with DMXAA resulted in a significant decrease in proliferation of WT and Parp7^KO^ cells compared with untreated cells. However, the decrease in proliferation was more evident in the Parp7^KO^ cells compared with WT cells, suggesting that PARP7 may regulate DMXAA-mediated suppression of cell proliferation by downregulation levels of IFN-β (Figure 2J). Because several members of the ARTD family have been implicated in type I IFN signaling [30], we evaluated whether these were affected by DMXAA and/or RBN-2397. We determined the expression levels of *Parp1*, *Parp2*, *Parp3*, *Parp4*, *Tnks1* (Parp5a), *Tnks2* (Parp5b), *Parp6*, *Parp7*, *Parp8*, *Parp9*, *Parp10*, *Parp11*, *Parp12*, *Parp13*, *Parp14,* and *Parp16* (Figure 2K). The graphs for the individual PARPs are provided in Appendix A. We observed that *Parp3*, *Parp6*, *Parp9*, *Parp10*, *Parp11*, *Parp12, Parp13,* and *Parp14* were significantly upregulated in the Parp7^KO^ cells. *Parp13* and *Parp14* were further upregulated in response to DMXAA. In WT cells, RBN treatment had little effect on the expression levels of different ARTD family members. However, DMXAA increased the expression of *Parp9*, *Parp10*, *Parp12*, *Parp13,* and *Parp14*. The levels *Parp13,* and *Parp14* were further increased after co-treated with RBN-2397.

### 3.2. Loss of PARP7 Increases Levels of Unphosphorylated ISGF3

To determine the mechanism by which PARP7 regulates type I IFN signaling, we analyzed the expression and phosphorylation levels of different proteins in the type I IFN signaling cascade in WT and Parp7^KO^ cells. As expected, PARP7 protein levels were stabilized by RBN-2397 but not by DMXAA (Figure 3A). The levels of cGAS were not affected by DMXAA or loss of PARP7 function. STING becomes ubiquitinated after activation, resulting in its rapid degradation [5]. In agreement with this, a higher molecular weight band was observed after 2 h DMXAA treatment. STING phosphorylation (S365) was evident after 2 h but was markedly reduced at later time points as would be predicted by its DMXAA-induced degradation. Neither the levels of native nor phosphorylated STING were affected by PARP7 loss or its inhibition. TBK1 phosphorylation (S172) was also detected after 2 h and sustained at 24 h after stimulation with DMXAA. RBN-2397 had no effect on pTBK1, and no decreases in TBK1 levels in response to loss of PARP7 function were observed. Phosphorylation of IRF3 (S396) was detected at 2 h but it was markedly reduced at later time points. pIRF3 levels trended towards, being higher in the Parp7^KO^ cells, but this increase was not consistent among experiments. Phosphorylation of STAT1 (Y701) peaked at 6 h but was notably lower at the same time point in the Parp7^KO^ compared with WT cells. The levels of native STAT1 increased in Parp7^KO^ cells, and this was independent of DMXAA. The levels of STAT2 increased after 6 and 24 h of DMXAA treatment in the WT cells, but higher levels were observed in Parp7^KO^ and these were independent of DMXAA. The same pattern was observed for IRF9. Treatment with RBN-2397 for 48 h resulted in STAT1, STAT2, and IRF9 levels mirroring those observed in the Parp7^KO^ cells (Appendix A). Increases in the levels of STAT1, STAT2, and IRF9, which together constitute ISGF3, were also observed in MEFs isolated from *Parp7^H532A^* mice (Appendix A). The quantification of STAT1 protein levels confirmed significantly increased protein levels in response to PARP7 loss (Figure 3B). However, when quantifying the levels of pSTAT1 relative to native STAT1 in response to treatment, loss of PARP7 results in significantly lower amounts of phosphorylation (Figure 3C). Long-term exposure to low levels of IFN-β results in increased levels of unphosphorylated ISGF3 [31,32,33]. Thus, a possible explanation for the observed increase in unphosphorylated STAT1 may be the increased levels of IFN-β secreted from the Parp7^KO^ cells.

To understand how loss of PARP7 affects ISGF3-signaling, cytoplasmic and nuclear fractions of cells treated with DMXAA (+/− RBN-2397) for 6 h were analyzed (Figure 3D). Most of the phosphorylated STAT1 (Y701) was in the nucleus. Native STAT1 was present in the cytoplasm in both untreated and treated cells and was increased Parp7^KO^ cells. STAT1 was only slightly visible in the nucleus in response to treatment in the WT cells, and increased levels were observed after PARP7 inhibition. In WT cells, STAT2 and IRF9 levels increased in both the cytoplasm and the nucleus in response to DMXAA. Loss of PARP7 resulted in increased STAT2 and IRF9 levels in both cellular compartments, but this was not affected by DMXAA. Taken together, these data indicate that loss of PARP7 increases STAT1, STAT2, and IRF9 levels, which make up the ISGF3 signaling complex, and this is not affected by DMXAA. This could indicate that loss of PARP7 leads to active ISGF3 signaling, even in the absence of STING activation.

### 3.3. Loss of PARP7 Increases the Levels of Stat1, Stat2, Irf9, and ISGF3 Target Genes

To investigate whether the expression levels of *Stat1*, *Stat2,* and *Irf9* were affected by the inhibition or loss of PARP7, EO771 WT cells were treated with RBN-2397 for 2, 6, and 24 h, and the relative levels of mRNA were compared to untreated Parp7^KO^ cells (Figure 3E–G). We observed significant increases in the expression of all three genes in Parp7^KO^ cells and in MEFs isolated from *Parp7^H532A^* mice (Appendix A). RBN-2397 treatment increased *Stat2* and *Irf9* levels after 24 h. These data suggest that the increased protein levels are due to the increased transcriptional regulation.

ISGF3 signaling occurs through phosphorylated ISGF3 (P-ISGF3), which consists of pSTAT1, pSTAT2 and IRF9, or through unphosphorylated ISGF3 (U-ISGF3) in which none of the proteins in the trimeric complex are phosphorylated. P-ISGF3 and U-ISGF3 bind DNA and regulate different target genes [31,32,33]. To investigate whether PARP7 loss or its inhibition results in differential expression of P-ISGF3 and U-ISGF3 target genes, cells were treated with DMXAA (+/− RBN-2397) for 6 h, and mRNA levels were measured with RT-qPCR. Expression of the P-ISGF3 target gene *Myd88* [31], was unaffected by PARP7 loss (Figure 3H). However, both *Adar* and *Irf1* [31] were increased in the Parp7^KO^ cells (Figure 3I,J), and *Irf1* also increased in response to RBN-2397. These results suggest that despite lower levels of pSTAT1 in Parp7^KO^ cells, the signaling pathway is enhanced. The U-ISGF3 target genes, *Isg15* and *Mx1* [31,32,33], and the ISG15-induced *Usp18* [34,35] were all significantly upregulated in response to both PARP7 loss or inhibition (Figure 3K–M), and this increase was greater than that observed for the P-ISGF3 target genes. These data show that U-ISGF3 signaling is upregulated in response to PARP7 loss, likely due to the elevated basal levels of IFN-β.

### 3.4. Loss of PARP7 Results in Increased Levels of Canonical NF-κB Subunits, and PARP7 Modifies RelA

Previous studies have reported that PARP7 negatively regulates the type I IFN signaling pathway by modifying and repressing TBK1 [12,15]. To confirm this, we treated WT and Parp7^KO^ cells with the TBK1 inhibitor, MRT67307, or with BX795, an inhibitor of TBK1, that also inhibits IκB kinase α (IKKα) and IKKβ, which both act upstream of the transcription factor NF-κB. Following the inhibition, cells were stimulated with DMXAA for 2 h. Treatment with MRT67307 decreased *Ifnb1* levels, but it did not fully block DMXAA-stimulated *Ifnb1* levels (Figure 4A). However, the effect of MRT67307 was stronger in the cells lacking PARP7, possibly suggesting increased baseline TBK1 activity in these cells. BX795 treatment completely blocked the expression of *Ifnb1* in all cases, suggesting that other signaling components than TBK1 may regulate transcription of *Ifnb1*. The *Ifnb1* promoter contains binding sites for IRF3 but also for NF-κB [4,36], and previous studies have reported that TBK1 is dispensable for STING-induced NF-κB responses [37]. Based on this, we examined whether the levels of the NF-κB subunits, p50 and RelA, were responsive to DMXAA and whether they were affected by PARP7 loss. We observed that both p50 and RelA are expressed in EO771 cells, and that the levels increased in WT cells in response to DMXAA (Figure 4B,C). The levels were higher in response to both PARP7 loss and its inhibition, regardless of DMXAA treatment. In contrast, mRNA levels of *Rela* were not affected by DMXAA or PARP7, suggesting that the changes in protein levels occur after translation (Figure 4D). Knockdown of TBK1 or RelA reduced respective protein levels (Figure 4E) and resulted in decreased expression levels of *Ifnb1* in both WT and Parp7^KO^ cells (Figure 4F). However, this decrease was more prominent in the Parp7^KO^ cells. There was a trend for a further decrease in *Ifnb1* levels after combined knockdown of both TBK1 and RelA, but this was not statistically significant compared to either knockdown alone. We next determined whether PARP7 interacted with and MARylated the NF-κB subunits p50 or RelA. We observed that PARP7, and the catalytically inactive PARP7^H532A^ mutant, co-immunoprecipitated with both p50 and RelA (Figure 4G). WT PARP7 MARylated itself in both conditions, and we observed that PARP7 MARylated RelA but not p50. Taken together, these data indicate that, in addition to TBK1, components of the NF-κB pathway are involved in regulation of *Ifnb1*, and these components may be regulated by PARP7.

### 3.5. Loss of Parp7 Prevents Mammary Tumor Growth in Catalytic Mutant Parp7^H532A^ Mice

To investigate the role of PARP7 loss on mammary tumor growth and antitumor immunity, we determined the ability of WT and Parp7^KO^ EO771 cells to develop tumors in immunodeficient NSG mice, and in syngeneic tumor studies using WT (*Parp7^+/+^*) and catalytic mutant *Parp7^H532A^* mice. No differences in tumor growth were observed in NSG mice injected with WT or Parp7^KO^ cells (Figure 5A,B). In immunocompetent mice, loss of PARP7 in the cancer cells significantly decreased tumor growth compared to WT cells in *Parp7^+/+^* mice (Figure 5C,D). However, injection of WT cells into *Parp7^H532A^* mice resulted in reduced tumor growth compared with WT cells or Parp7^KO^ injected into *Parp7^+/+^* mice. Injection of Parp7^KO^ cells into *Parp7^H532A^* mice failed to grow tumors after 30 days, and those that developed regressed (Figure 5D,E). Similar findings were observed when injecting *Parp7^H532A^* mice with two additional EO771 Parp7^KO^ clones (Appendix A). *Ifnb1* and *Cxcl10* levels were increased in response to PARP7 loss in the tumors (Figure 5F). Since these cytokines are known to attract immune cells, we stained tumor sections with antibodies against T cells (CD3) or macrophages (CD68) (Figure 5G,I). The levels of tumor infiltrating T cells were significantly higher in the tumors arising from the Parp7^KO^ cells, while there was no significant difference between the recipient genotype (Figure 5H). Interestingly, the opposite was observed for tumor infiltrating macrophages, where loss of functional PARP7 in the recipient mice resulted in significantly increased levels of macrophages; however, this was independent of PARP7 expression in the cancer cells (Figure 5J). Taken together, these data show that loss of PARP7 in the tumors results in increased type I IFN signaling and subsequently increased T cell infiltration, while loss of PARP7 function in the recipient mice increased macrophage infiltration. The combined effect of PARP7 loss or its activity in both cancer cells and the immune system, impairs tumor growth.

### 3.6. Loss of PARP7 Activity Increases IFN Signaling in CD8^+^ T Cells and Increases M1 Macrophage Signaling

Since previous studies attribute the antitumor effect of PARP7 inhibition to CD8^+^ T cells [12,20], we determined if loss of PARP7 function affected IFN signaling in CD8^+^ T cells isolated from the spleens of *Parp7^+/+^* or *Parp7^H532A^* mice. To verify CD8^+^ T-cell isolation, relative levels of *Cd8a* mRNA were measured in the splenocytes prior to the isolation and in the resulting CD8^+^ T cells (Figure 6A). We observed an approximate 10-fold increase in *Cd8a* expression, confirming a successful isolation and enrichment of CD8^+^ T cells [38]. The isolated CD8^+^ T cells were either kept in a resting state or activated with CD3/CD28 beads prior to treatment with DMXAA for 24 h, and relative expression levels of type I, II, and III IFNs were determined. Loss of PARP7 activity significantly increased *Ifnb1* levels in both resting and activated CD8^+^ T cells (Figure 6B). This was further confirmed with significantly increased levels of secreted IFN-β (Figure 6C). DMXAA significantly increased type II IFN, *Ifng,* expression but this was independent of PARP7 (Figure 6D). In contrast, loss of PARP7 activity significantly increased type III IFN, *Ifnl2*, expression in activated CD8^+^ T cells (Figure 6E). Collectively, these data underline PARP7′s repressive role in type I and type III but not type II IFN signaling in cytotoxic CD8^+^ T cells.

Since we observed an increase in tumor infiltrating macrophages in the *Parp7^H532A^* mice (Figure 5I,J), we determined whether loss of PARP7 activity affected M0, M1, and M2 macrophages in vitro. BMDMs were harvested from *Parp7^+/+^* and *Parp7^H532A^* mice and mRNA expression levels of phenotypic markers were determined after polarization. Loss of PARP7 activity significantly increased levels of the M1 marker *Il6* (Figure 6F) in the proinflammatory M1 macrophages. In addition, the levels of *Ifnb1* were upregulated in BMDMs from *Parp7^H532A^* mice (Figure 6G). The M2 phenotypic marker, *Arg1*, was slightly upregulated in the *Parp7^H532A^* M2 macrophages, but this increase was not as pronounced as that for *Il6* and *Ifnb1* in the M1 phenotype (Figure 6H). Taken together, these results suggest that the *Parp7^H532A^* mice have increased proinflammatory M1 macrophage activity, which further underlines PARP7′s importance in antitumor immunity.

## 4. Discussion

Here, we show that loss of PARP7 increases IFN-β levels and downstream U-ISGF3 signaling in EO771 murine mammary cancer cells. PARP7 loss or inhibition moderately reduced cell proliferation but did not affect tumor growth in immunodeficient mice. However, loss of PARP7 catalytic activity in recipient mice injected with WT or Parp7^KO^ cells resulted in a greater reduction in tumor growth compared with WT mice. Our results show that PARP7 loss increases type I IFN and downstream signaling pathways leading to increased tumor targeting by the immune system.

In agreement with previous studies, PARP7 loss or its inhibition increased *Ifnb1* mRNA and secreted IFN-β protein levels [15]. Constitutive *Cxcl10* levels were higher in the absence of PARP7 but lower after DMXAA stimulation compared with DMXAA + RBN-2397-treated WT cells. This is consistent with RBN-2397 causing a more effective reduction in tumor growth compared with Parp7^KO^ because of higher *Cxcl10* levels [12]. It was suggested that this was because of RBN-2397′s “trapping” of PARP7, altering its function [12], which would not occur in Parp7^KO^ cells. Another explanation could be off-target effects of RBN-2397, which at the doses used in our study is known to inhibit other ARTDs, including PARP1, PARP2, and PARP12 [19,20]. Loss of PARP7 also increased levels of *Pdl1*, which acts to repress immune responses against cancer cells [4]. Thus, combining PARP7 inhibition with immune therapy may be more effective than targeting PARP7 alone. In this regard, it will be interesting to follow the current RBN-2397 combined pembrolizumab clinical trial (NCT05127590). Further, we observed an increase in the expression levels of *Parp14* in the cells deficient in PARP7, and this was further increased following DMXAA treatment. Previous studies demonstrated that PARP14 is induced by IFN-β, and depletion of PARP14 was showed to decrease *Ifnb1* expression [39]. Thus, the increased expression levels are likely a result of the elevated type I IFN signaling observed in the Parp7^KO^ cells. We cannot exclude that the elevated levels of *Parp14* contribute to some of the phenotypes observed in the Parp7^KO^ cells.

ISGF3 functions in its phosphorylated, P-ISGF3, or unphosphorylated, U-ISGF3, forms. Both P-ISGF3 and U-ISGF3 regulate constitutive ISGs, with U-ISGF3 being sufficient to maintain and prolong antiviral immunity [40]. PARP7 loss increased constitutive levels of STAT1, STAT2, and IRF9, which form the ISGF3 signaling complex. The increase in ISGF3 is not unique in the EO771 Parp7^KO^ cells, since this was also observed in MEFs isolated from *Parp7^H532A^* mice. RBN-2397 and KMR-206 increase STAT1 levels in CT26 cells, but whether STAT2 and IRF9 are also increased is not known [12]. Although we observed an increase P-ISGF3 regulated genes in the Parp7^KO^ cells, the difference was greater for U-ISGF3 regulated genes, *Isg15*, *Mx1*, and *Usp18*. ISG15 is a small ubiquitin-like protein that increases the levels of USP18, a negative regulator of type I IFN signaling [34,35]. USP18 inhibits type I IFN signaling by binding to the intracellular domains of IFNAR2, preventing the phosphorylation of STAT1 and STAT2 [35]. The increased *Usp18* levels might explain the decrease in pSTAT1 that we observed in Parp7^KO^ cells compared with PARP7 inhibition in EO771, which has also been reported in CT26 cells [12,19].

PARP7 inhibition-induced repression of type I IFN signaling requires STING and involves the MARylation and inactivation of TBK1 [12,15]. The importance of TBK1 was confirmed by pharmacological inhibition. Interestingly, TBK1 inhibition with MRT67307 only partially inhibited DMXAA stimulated *Ifnb1* levels after PARP7 loss. Complete inhibition was only observed after treatment with the dual TBK1 and IKK inhibitor, BX795, suggesting the involvement of NF-κB signaling. In agreement with these findings, the levels of p50 and RelA were upregulated in the cells lacking PARP7 or following PARP7 inhibition. The same was not observed for *Rela* mRNA levels, suggesting that this upregulation occurs post-translationally. In line with this, we observed that RelA but not p50 was MARylated by PARP7. However, the importance of RelA in PARP7 inhibitor dependent tumor regression is not known and represents an important for future research. Taken together, our data reveal that, like PARP1 and PARP10, PARP7 also regulates NF-κB signaling and uses this mechanism to modulate type I IFN signaling [41].

For some cancer cells, like NCI-H1373, PARP7 inhibition decreases viability in the absence of immune cells, whereas for others, like CT26 cells, antitumor effects of PARP7 inhibition or PARP7 loss requires an intact immune system [12]. CRISPR interference and activation screens identified that inactivation of the *AHR* gene is associated with resistance to PARP7 inhibition [12,21]. Since AHR ligand modulation alters PARP7 levels and cellular responses to PARP7 inhibition, potential inhibition of AHR and PARP7 combination therapy has been proposed [21]. The EO771 cells used in our study do not express AHR, and similar to CT26 cells, they were relatively insensitive to RBN-2397-induced decreases in cell proliferation, and PARP7 loss had no effect on tumor growth in immunodeficient NSG mice. However, loss of PARP7 in cancer cells significantly decreased tumor growth in recipient mice with an intact immune system. *Parp7^H532A^* mice injected Parp7^KO^ cells failed to develop tumors, and the ones that did regressed. These data show that lack of AHR expression in cancer cells did not affect the Parp7^KO^-dependent antitumor immunity in vivo. This suggests that even though some cancer cells may be insensitive to the antiproliferative effects of PARP7 inhibition in the absence of immune cells, they may still be sensitive to antitumor immunity in the presence of immune cells.

Tumor secretion of IFN-β can activate dendritic cells [42] but also promote the proliferation and activation of CD8^+^ T cells [43,44]. Increased levels of tumor infiltrating T cells were evident in tumors arising from Parp7^KO^ cells, while increased levels of tumor infiltrating CD68^+^ macrophages were observed in tumors from the *Parp7^H532A^* recipient mice. Moreover, CD8^+^ T cells isolated from *Parp7^H532A^* mice had low levels of *Ifnb1* and *Ifnl2*, but they were highly induced after DMXAA-mediated STING activation. Loss of PARP7 activity resulted in increased *Il6* and *Ifnb1* levels in M1 macrophages in vitro. These results suggest that *Parp7^H532A^* mice harbor hyperactive proinflammatory macrophages, resulting in increased tumor infiltration, and reduced tumor growth. Thus, releasing the brake in the form of PARP7 loss or its inhibition, combined with pressing the accelerator, in the form of STING activation, hyperactivates IFN responses in both tumor and immune cells, thereby priming the immune system to attack and target cancer cells.

Acute exposure to high levels of IFN-β induces antitumor responses such as extensive DNA damage, resulting in cytotoxicity. In contrast, chronic exposure to low levels of IFN-β can promote tumor growth and contribute to DNA damage resistance through U-ISGF3-dependent regulation of IFN-related DNA damage resistance signature (IRDS) [31]. Since high levels of U-STAT1, U-STAT2, and IRF9 are needed for the expression of the IRDS genes, the observed increases in U-ISGF3 we observed in Parp7^KO^ cells would enable them to survive endogenous and therapy-induced DNA damage [45]. However, tumor growth was reduced when Parp7^KO^ cells were injected into *Parp7^+/+^* and in *Parp7^H532A^* mice. This suggests that even though PARP7 loss increases U-ISGF3 signaling, the increased STING-dependent IFN-β response and activated antitumor immunity dominate over the protumorigenic actions of U-ISGF3. Our proposed mechanism of action based on the findings in this study is summarized in Figure 7.

## 5. Conclusions

To better understand the role of PARP7 inhibition in breast cancer and antitumor immunity, we characterized the effect of PARP7 loss on type I IFN signaling in EO771 cells and on tumor growth in syngeneic models using catalytically deficient *Parp7^H532A^* mice. Although CRISPR/Cas9-mediated gene editing is an effective tool to study protein function, it is susceptible to off-target effects, clonal selection, and cell signaling adaptation due to the loss of target protein expression [46,47]. However, we observed good agreement among Parp7^KO^ clones and between PARP7 loss and pharmacological inhibition of PARP7 with RBN-2397 in vitro, suggesting that the increases in type I IFN were not due to unwanted errors introduced by gene editing. We show that PARP7 loss or inhibition induced type I IFN signaling, and this induction was further enhanced with STING activation in EO771 cells. Using preclinical tumor models, Parp7^KO^ cells combined with *Parp7^H532A^* mice, we found that loss of PARP7 activity in recipient mice resulted in smaller tumors compared with *Parp7^+/+^* mice, while tumors failed to develop in *Parp7^H532A^* mice injected with Parp7^KO^ cells. Although our strategy allowed us to distinguish between the effects of PARP7 loss in cancer cells versus recipient mouse, we cannot exclude potential confounding effects resulting from the use of gene edited Parp7^KO^ cells which may differ from PARP7 inhibition. Regardless of these limitations, our results confirm the role of PARP7 as a repressor of type I IFN signaling but also show the importance of targeting PARP7 activity in immune cells to improve antitumor immunity.

## Figures and Tables

**Figure 1 cancers-15-03689-f001:**
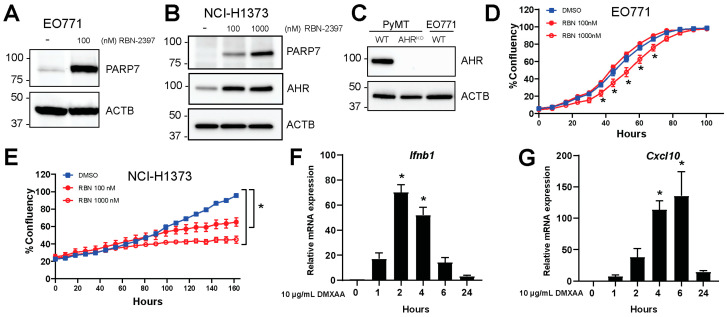
EO771 cells express PARP7 and are responsive to DMXAA stimulated increases in type I interferon (IFN) signaling. (**A**) EO771 or (**B**) NCI-H1373 cells were treated with RBN-2397 for 24 h to inhibit PARP7 activity and enable visualization with Western blotting. Levels of AHR were also determined. (**C**) AHR levels in WT and Ahr^KO^ PyMT cells and EO771 cells were determined with Western blotting. (**D**) RBN-2397 treatment slightly but significantly reduces EO771 cell proliferation, whereas (**E**) NCI-H1373 cells are sensitive to the antiproliferative effects of RBN-2397. Cell proliferation was analyzed in an IncuCyte instrument. (**F**,**G**) EO771 cells are responsive to stimulation with the STING activator, DMXAA. Cells were treated with DMXAA for 1, 2, 4, 6, and 24 h, and relative (**F**) *Ifnb1* and (**G**) *Cxcl10* mRNA levels were analyzed by RT-qPCR. * Denotes statistical significance (*p* < 0.05) from the DMSO treated samples (**D**,**E**) or untreated samples (0 h) (**F**,**G**). Original western blots has been presented in Appendix A.

**Figure 2 cancers-15-03689-f002:**
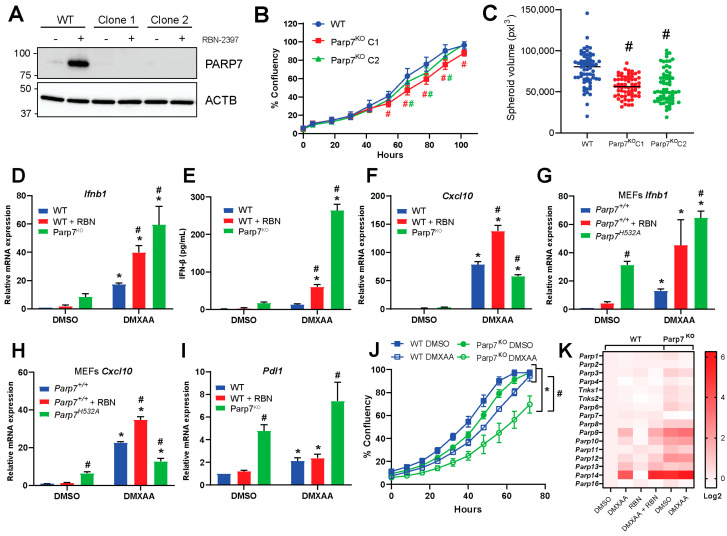
PARP7 loss or its inhibition increases type I IFN signaling. (**A**) PARP7 protein levels were determined in EO771 WT cells and Parp7^KO^ clone 1 and 2 by Western blotting. Only the WT displays a PARP7 band in response to PARP7 inhibition. (**B**) Both Parp7^KO^ clones proliferate significantly slower than the WT cells. (**C**) EO771 Parp7^KO^ clones 1 and 2 form significantly smaller spheroids than the WT cells. (**D**) *Ifnb1* mRNA levels increased in EO771 Parp7^KO^ and RBN-2397-treated WT cells. Cells were treated with 10 µg/mL of DMXAA (+/− 100 nM RBN-2397) for 2 h, and relative mRNA levels were measured with RT-qPCR. (**E**) IFN-β protein levels are higher after 4 h treatment of WT cells with DMXAA +/− RBN-2397 and DMXAA in EO771 Parp7^KO^ cells. (**F**) *Cxcl10* mRNA levels increased after 6 h DMXAA treatment in WT and Parp7^KO^ cells, with higher levels in WT cells co-treated with DMXAA and RBN-2397 compared with DMXAA-treated Parp7^KO^ cells. (**G**) *Ifnb1* levels increased in both untreated and DMXAA-treated *Parp7^H532A^* MEFs, and in WT MEFs co-treated with DMXAA and RBN-2397. (**H**) *Cxcl10* mRNA levels increased after 6 h treatment with DMXAA *Parp7^H532A^* and WT MEFs, with higher levels in WT MEFs co-treated with DMXAA + RBN-2397 compared with DMXAA *Parp7^H532A^* MEFs. (**I**) Parp7^KO^ cells display elevated constitutive *Pdl1* levels compared with WT cells treated +/− DMXAA. (**J**) DMXAA treatment results in a significant decrease in proliferation in the EO771 WT and Parp7^KO^ cells, with greater effect observed in Parp7^KO^ cells. (**K**) Heatmap showing expression levels of different PARPs in response to DMXAA and/or RBN-2397 in EO771 WT and Parp7^KO^ cells. Cells were treated with 10 µg/mL of DMXAA (+/− 100 nM RBN-2397) for 2 h, and relative mRNA levels were measured with RT-qPCR. * Denotes statistical significance (*p* < 0.05) compared to DMSO; # (for (**B**) these are colored to indicate which graph they correspond to) denotes statistical significance due to loss the of PARP7 activity. Original western blots has been presented in Appendix A.

**Figure 3 cancers-15-03689-f003:**
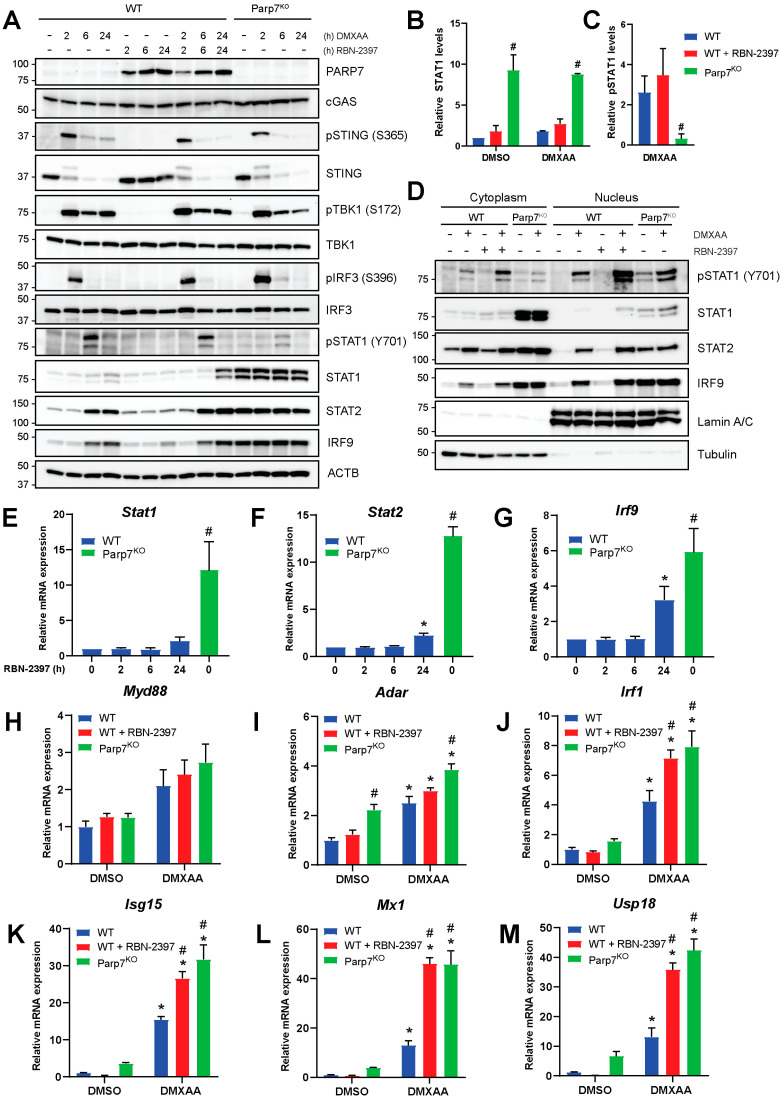
Levels of unphosphorylated interferon stimulated gene factor 3 (ISGF3) are upregulated in EO771 Parp7^KO^ cells. (**A**) EO771 WT and Parp7^KO^ cells have intact STING regulated type I IFN signaling pathway. The expression and/or phosphorylation of cGAS, STING, TBK1 and IRF3 are similar between WT and Parp7^KO^ cells. However, Parp7^KO^ cells have increased levels of unphosphorylated signal transducer and activator of transcription 1 (STAT1), STAT2, and interferon regulatory factor 9 (IRF9). Cells were treated with 10 µg/mL of DMXAA (+/− 100 nM RBN-2397) for 0, 2, 6, or 24 h, and samples were subjected to SDS-PAGE and Western blotting. (**B**) Levels of native STAT1 are higher in the EO771 Parp7^KO^ cells independent of DMXAA treatment. (**C**) Levels of phosphorylated STAT1 after DMXAA treatment relative to native STAT1 are lower in the EO771 Parp7^KO^ cells compared with both WT and RBN-2397-treated WT cells. (**D**) EO771 Parp7^KO^ cells have increased cytoplasmic and nuclear levels of STAT1, STAT2, and IRF9 in both untreated and treated samples. (**E**) *Stat1*, (**F**) *Stat2,* and (**G**) *Irf9* mRNA levels are higher in EO771 Parp7^KO^ cells and after 24 h RBN-2397 treatment in WT cells. (**H**) Expression levels of P-ISGF3 target genes *Myd88* is not significantly affected by PARP7 loss, while target genes (**I**) *Adar* and (**J**) *Irf1* are slightly higher in the EO771 Parp7^KO^ cells. (**G**) Expression levels of U-ISGF3 target genes, (**K**) *Isg15* and (**L**) *Mx1*, and (**M**) ISG15-induced gene *Usp18* are all significantly higher in EO771 Parp7^KO^ cells and RBN-2397-treated WT cells. Cells were treated with 10 µg/mL of DMXAA (+/− 100 nM RBN-2397) for 6 h, and mRNA levels were quantified with RT-qPCR. * Denotes statistical significance (*p* < 0.05) compared to no treatment (**E**–**G**) or DMSO (**H**–**M**); # denotes statistical significance due to the loss of PARP7. Original western blots have been presented in Appendix A.

**Figure 4 cancers-15-03689-f004:**
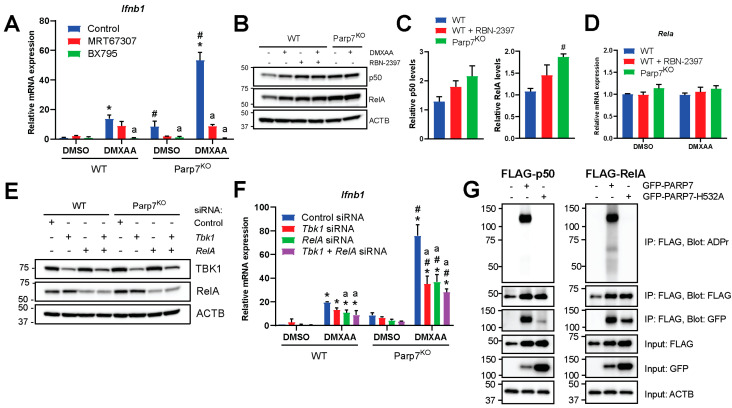
Nuclear factor κB (NF-κB) is involved in the increased type I IFN signaling observed in EO771 Parp7^KO^ cells. (**A**) Inhibition of TANK-binding kinase 1 (TBK1) with MRT67307 significantly decreases *Ifnb1* mRNA levels in EO771 Parp7^KO^ cells but not in WT cells. Inhibition of both TBK1 and the IκB kinases (IKKs) with BX795 blocks *Ifnb1* mRNA expression in all cases. Cells were treated with either 1 µM of MRT67307 or 100 nM of BX795 for 24 h, followed by 10 µg/mL of DMXAA for 2 h. (**B**) Treatment with DMXAA and/or RBN-2397 increases protein levels of both p50 and RelA in EO771 WT cells while levels are higher in Parp7^KO^ cells in both untreated and treated cells. Cells were treated with 10 µg/mL of DMXAA (+/− 100 nM of RBN-2397) for 24 h and proteins were visualized by Western blotting. (**C**) Levels of expressed NF-κB subunits are higher in response to PARP7 inhibition, and levels of RelA are significantly higher in the Parp7^KO^ cells. Protein levels of p50 and RelA in untreated samples were quantified by normalizing to the loading control. (**D**) Expression levels of *Rela* mRNA are not affected by DMXAA or PARP7. (**E**) Knockdown of either TBK1, RelA, or both, in EO771 WT and Parp7^KO^ cells. After 48 h of transfection with 25 nM of siRNA targeting *Tbk1*, *Rela*, or both, protein levels were visualized by Western blotting. (**F**) Knockdown of TBK1 and RelA results in significantly decreased *Ifnb1* mRNA levels in Parp7^KO^ cells but not in WT cells. After 48 h of transfection with 25 nM of siRNA for *Tbk1*, *Rela*, or both, cells were treated with 10 µg/mL of DMXAA for 2 h. (**G**) p50 and RelA interact with WT and catalytically inactive PARP7, and RelA is MARylated by WT PARP7. Cos-1 cells were transfected with FLAG-tagged p50 or RelA together with either GFP-tagged WT PARP7 or the catalytically inactive H532A mutant. The FLAG-tagged p50 or RelA was immunoprecipitated, and the interaction and modification was examined by Western blotting. * Denotes statistical significance (*p* < 0.05) compared to DMSO; # denotes statistical difference due to the loss of PARP7; “a” denotes statistical difference compared to no inhibitor (**A**) or to the nontargeting siRNA (**F**). Original western blots have been presented in Appendix A.

**Figure 5 cancers-15-03689-f005:**
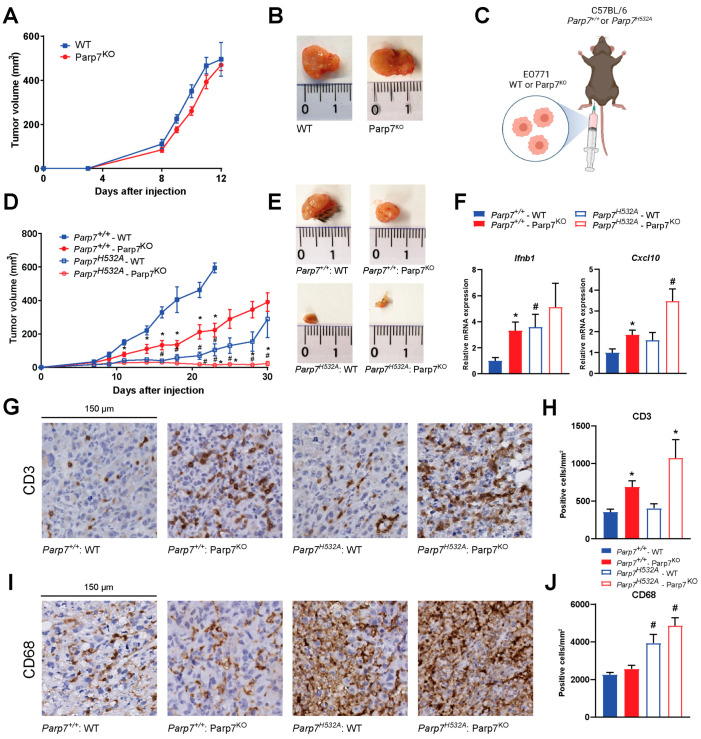
Loss of PARP7 prevents mammary tumor growth in immunocompetent mice. (**A**) No difference in tumor growth is observed between WT and Parp7^KO^ cells injected into immunodeficient NSG-mice, *n* = 8. (**B**) Representative images of tumors dissected from NSG mice injected with WT (left) or Parp7^KO^ (right) cells. (**C**) Graphical representation of experimental setup indicating injection of either WT or Parp7^KO^ cells into either C57BL/6 *Parp7^+/+^* or *Parp7^H532A^* mice. Figure made with BioRender. (**D**) Loss of PARP7 in the cancer cells significantly decreases tumor growth in *Parp7^+/+^* mice, and this is further decreased in *Parp7^H532A^* mice, *n* = 6–12. * Denotes statistical significance (*p* < 0.05) between the genotype of the cells injected, while # denotes statistical significance between the genotype of the mice. (**E**) Representative images of tumors dissected from *Parp7^+/+^* mice injected with WT or Parp7^KO^ cells and from *Parp7^H532A^* mice injected with WT or Parp7^KO^ cells, respectively. (**F**) Tumor expression levels of *Ifnb1* and *Cxcl10* are increased in response to loss of PARP7 activity in either cancer cells or in recipient immune system. * Denotes statistical significance (*p* < 0.05) compared to the WT cells injected in the corresponding mouse strain, and # denotes statistical significance compared to corresponding cell line injected in the *Parp7^+/+^* mice. (**G**) Injection of Parp7^KO^ cells increases tumor infiltration of T cells. Representative images of tumor sections stained with antibody against CD3. (**H**) Quantification of T-cell infiltration with CD3-positive cells per mm^2^, *n* = 8. (**I**) Loss of functional PARP7 in recipient mice results in increased tumor infiltration of macrophages. Representative images of tumor sections stained with antibody against CD68. (**J**) Quantification of macrophage infiltration, *n* = 8. For (**H**,**J**), # denotes statistical significance (*p* < 0.05) between the genotype of the mice injected, while * denotes significance between the cells injected.

**Figure 6 cancers-15-03689-f006:**
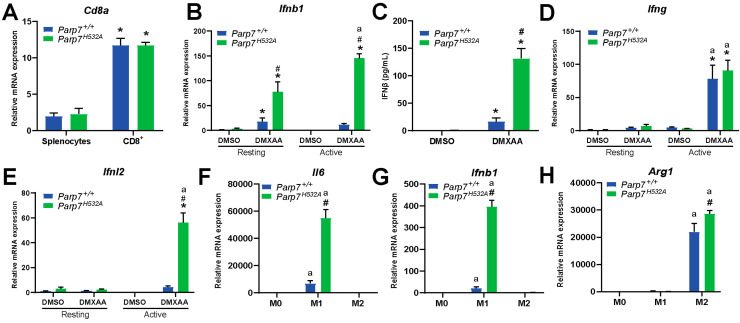
Loss of PARP7 increases type I IFN signaling in CD8^+^ T cells and increases M1 macrophage signaling. (**A**) Enrichment of *Cd8a* mRNA verifies CD8^+^ T-cell isolation. Levels of *Cd8a* mRNA expression is increased in isolated CD8^+^ T cells compared to the splenocyte starting material. Expression levels were measured with RT-qPCR. (**B**) Levels of type I IFN *Ifnb1* are significantly increased in both resting and activated CD8^+^ T cells isolated from *Parp7^H532A^* mice. Cells were either kept in a resting state or activated with CD3/CD28 beads prior to treatment with 10 µg/mL of DMXAA for 24 h. (**C**) Loss of PARP7 function results in higher secreted levels of IFN-β from activated CD8^+^ T cells treated with 10 µg/mL of DMXAA for 24 h. (**D**) The levels of the type II IFN *Ifng* do not significantly differ between the genotypes, while (**E**) the levels of the type III IFN *Ifnl2* are significantly upregulated in activated CD8^+^ T cells isolated from *Parp7^H532A^* mice. (**F**) Levels of *Il6* and (**G**) *Ifnb1* are higher in M1 macrophages isolated from *Parp7^H532A^* mice compared with *Parp7^+/+^* mice. (**H**) M2 macrophages from *Parp7^H532A^* mice have higher expression levels of *Arg1*. For (**F**–**H**), bone-marrow-derived macrophages were isolated and polarized, and the mRNA levels were determined with RT-qPCR and normalized to the M0. * Denotes statistical significance (*p* < 0.05) compared to splenocytes (**A**) or DMSO (**B**–**E**); # denotes statistical significance due to loss of PARP7 function (**B**,**C**,**E**–**H**); “a” denotes statistical significance compared to the resting cells (**B**,**D**,**E**) or the unpolarized M0 macrophages (**F**–**H**). *n* = 3.

**Figure 7 cancers-15-03689-f007:**
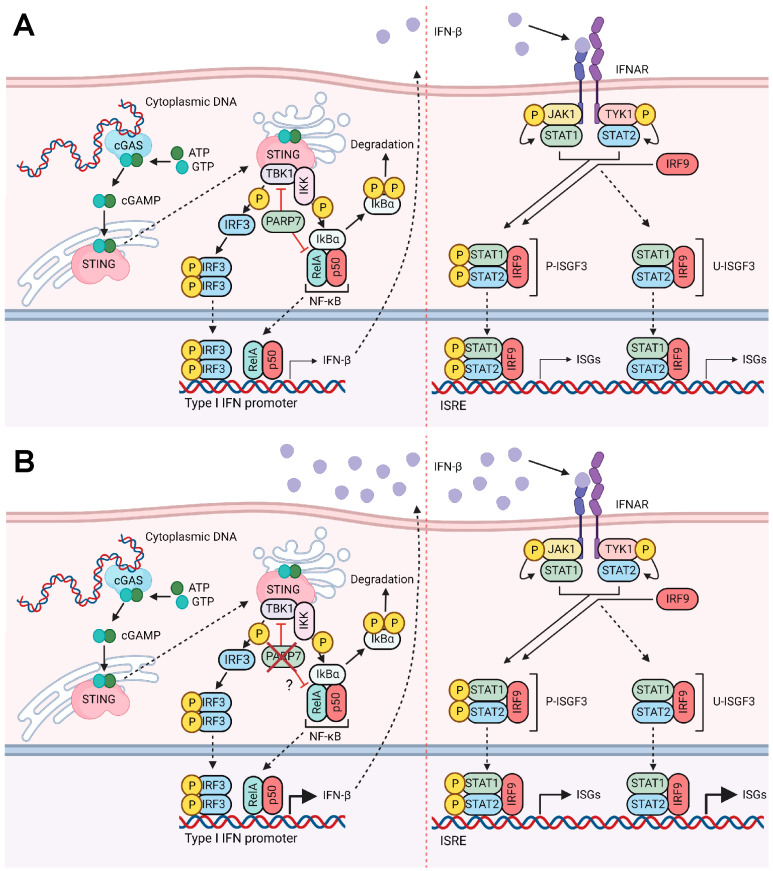
Proposed mechanism of action. (**A**) In the presence of cytoplasmic DNA, cGAS is activated and catalyzes the synthesis of cGAMP, which activates STING. STING subsequently activates TBK1, which phosphorylates and activates IRF3. STING activation also results in the release of NF-κB from IκBα. Both pIRF3 and NF-κB translocate to the nucleus and bind to the promoter region of type I IFNs, such as *Ifnb1*. PARP7 inhibits the activity of both TBK1 and the RelA subunit of NF-κB. Secreted IFN-β binds to IFNAR, which results in phosphorylation of STAT1 and STAT2. Together with IRF9, they form P-ISGF3 which regulates the expression of ISGs. Upon sustained IFN-β, unphosphorylated ISGF3 is upregulated, which results in transcription of certain ISGs. (**B**) Loss of PARP7 results in increased production and secretion of IFN-β, which further increases both P-ISGF3 and U-ISGF3 signaling. Figure made in BioRender (license agreement YX25IUCM4M to J.M.).

## Data Availability

All data are included in the paper. There are no databases associated with this manuscript.

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
