# Peer review of "Loss of PARP7 Increases Type I Interferon Signaling in EO771 Breast Cancer Cells and Prevents Mammary Tumor Growth by Increasing Antitumor Immunity"

_cancers, 2023, doi:10.3390/cancers15143689_

Round 1
Reviewer 1 Report
This is quite interesting and very wellwritten. I have just one comment on the experimental design. In Figure 6F-H, the authors use BMDM from PARP7 WT and H532A mice to show that the H532A mice preferentially use M1 proinflammatory macrophages, which further contributes to antitumor immunity. Macrophages are a highly plastic population and their phenotypes will be rapidly altered by the tumor microenvironment. A better experiment would be to study macrophage phenotypes in the tumor, using IF/IHC to quantify the co-localization of M1 markers with IFN-I signaling activation markers, or by using flow to identify their co-expression.
Reviewer 2 Report
In this study, the authors investigate how PARP7 loss leads to decreased breast cancer growth, and study both loss of PARP7 in the tumour cells as well as in the surrounding tissue/immune system. They do this with an elegant mouse model as well as in vitro cell models, with mostly RT-qPCR and western blots to analyse influence of PARP7 loss on type I interferon signalling. Part of the work confirms the effect of PARP7KO on IFN signalling as studied in other systems. The work is overall well executed and described and my suggestions below are mainly intended to improve the structure of the manuscript and to clarify some unclear passages.
Line 85 I would suggest to not state that Phthal01 is a specific PARP7 inhibitor, as in a publication from the Cohen lab it clearly also inhibits PARP10 (Rodriguez, eLife 2021).
In the methods section, some details are missing. As an example, I could not find in reference 24 how the mentioned plasmids (pEGFP-PARP7 and mutant) were generated. Chasing references in the methods section is a nuisance to many researchers looking to reproduce findings, and I’d therefore encourage the authors to include experimental details where possible. The “lab generated PARP7” antibody [22] could also be briefly described: is it mono- or polyclonal, bleed or purified, etc. 2.11 “RT-qPCR was performed as described above”. Above reads: “RT-qPCR was set up as previously described [22]”.
Line 303 I suppose it is reasonable that the authors refer to a model they are developing and intend to publish later, however, the abbreviation “MMTV-PyMT” should be explained at this point.
Figure 2F-H: it would have been interesting to test the RBN compound on the PARP7 mutant cells, to determine whether it is solely acting by blocking PARP7 activity. It is always questionable how specific inhibitors are, and if the inhibitor has no effect whatsoever in the PARP7KO cells, this would be a nice indicator. The authors rightly discuss inhibitor specificity in the discussion.
Figure 4G: which antibody did the authors use to detect ADPr? It is my understanding that the available antibodies can have quite diverse specificities. Also, as the authors PARP7 knockouts at their disposal, as well as an inhibitor, did you try to IP endogenous RelA to determine whether the endogenous protein is modified? This modification would then be absent in the PARP7 KO or inactive mutant cells.
Overall, the authors are investigating the role of PARP7 in a complex signalling network: it is highly interesting to distinguish between the clinical outcome of inhibition of PARP7 in tumour cells, or inhibition of PARP7 in tissue surrounding the tumour. It would be very helpful for the reader if the authors could provide a scheme of the relevant pathway and indicate where PARP7 comes in.
On my printout, also the labelling of KO cells is hard to read, it is for example impossible to distinguish PARP7KO and PARP7H532A
Reviewer 3 Report
Paper from Rasmussen et al. introduces a novel and interesting concept that the combined effects of PARP7 loss or its activity in both cancer cells and the immune system impairs tumor growth. However, the title seems a little too broad when there is only one single AhR-free cell line EO771 is presented in the paper. Also I would like to know the number of mice used as representatives for figures 5G-J and 6A-H. Other concerns are:
(1) Fig. 2D shows slightly cell growth reduction of EO771 in the presence of high conc. Of RBN2397, while Fig. 3B shows no significant cell growth difference between WT and PARP7 KO, and then leaves behind a speculation of RBN2397 function. The paper could be easily enhanced by including RBN2397 condition in Fig. 3B.
(2) There are dramatic growth changes between WT and PARP7 KO cells shown in Fig. 3C and 3F. PARP7 KO cells are from selected clone. To avoid artifacts, the RBN2397 condition should be included in these assays for WT and PARP7 KO cells; stable PARP7 expression (with vector alone as control) could be re-introduced into the PARP7 KO for these assays too.
(3) Levels of Stat1, State2, Irf9 and ISGF3 are shown to be elevated in PARP7 KO cells compared to WT cells. Would a longer RBN2397 treatment of WT cells have the same effect? Are these levels back down when PARP7 expression is re-introduced into the PARP7 KO cells?
Reviewer 4 Report
Cancers-2441502
Loss of PARP7 prevents mammary tumor growth by 2 increasing type I interferon signaling
In the manuscript, Rasmussen and colleagues analyze the effect of PARP7 loss on breast cancer tumor development using both in vitro mouse mammary cancer cells and in vivo preclinical syngeneic tumor models. The authors show that PARP7 loss is associated with activation of the type I IFN signaling, i.e. activation of interferon-stimulated gene factor 3 (ISGF3) and specifically unphosphorylated-ISGF3 regulated target genes. Importantly, the authors demonstrate that PARP7 loss has a clear effect on tumor growth in immunocompetent mice models, therefore highlighting the key role of PARP7 in immune cells.
The experiments are well-performed, and the data are clearly shown, although sometimes the flow of the text is not easy to follow; for this reason, I suggest to include some schemes to facilitate the overall reading. In general, I believe that the paper is suitable for publication in Cancers; however, some additional experiments are required to better demonstrate/clarify specific aspects.
Minor points:
In the Material and Methods section, please include the antibody used to detect ADP-ribosylation.
Line 306: please, indicate the percentage of observed decreased proliferation in EO771 cells upon RBN-2397 treatment.
Figure 1
- please show levels of ARH in NCI-H1373, to correlate the effects of PARP7 inhibition on cell proliferation;
- levels of PARP7 in NCI-H1373 are almost undetectable compared to EO771, while the effect of RBN-2397 on NCI-H1373 is quite evident when compared to EO771 cells. Can the author exclude that the effect mediated by the inhibitor is not occurring through additional PARPs (ex PARP1/2 or PARP12)?
I suggest checking levels and activities of other PARP enzymes to exclude the contribution of additional PARPs in the phenotype, considering the involvement of multiple PARPs in the IFN and NfKB signaling pathways.
Figure 2A: from the WB, both clones 1 and 2 show a residual signal of PARP7, while the effect observed with the inhibitor RBN-2397 is very strong. Could the author comment on that?
Figure 2 B,C: PARP7 KO clones did not show effects on cell proliferation, while the show a defect in the formation of spheroids. Can the author comment on this difference?
Figure 2D-H: can the author clarify why the combined PARP7 inhibition and DMXAA treatment show different effect on Ifnb1 and Cxcl10?
Figure 3: what about levels of STING in PARP7 ko cells treated with PARP7 inhibitor?
Similarly, the authors show that increased levels of STAT1-2 and IRF9 are independent on PARP7 catalytic activity, while depending on PARP7 loss.
Please, include the effect of PARP7 inhibitor on PARP7 ko cell.
Line 416: can the author show immunofluorescence pictures of the pSTAT1?
What about pSTAT2 levels?
Figure 4G: the authors clearly show a co-Ip between PARP7 and p50 and RelA. Can the author comment on the reduced interaction with p50 and PARP7H532A?
Moreover, can the author demonstrate this result and subcellular localization of over-expressed PARP7 in the EO771 cell line?
Please, confirm ADP-ribosylation of RelA in in vitro assays using recombinant purified proteins.
Major points
A link between ADP-ribosylation of RelA and the tumor regression is completely missing. Can the authors perform some experiments to address this point? This would further increase the interest of the findings.
Round 2
Reviewer 3 Report
Since the re-introduction of PARP7 to cells usually takes a longer time, be it lentiviral (authors indicated that their department does not allow this practice there) or gene knocking in, Rasmussen et al have satisfied most of my concerns for which the experiments could be done in a relatively short time. However, it seems that RBN2397 condition (maybe needs 48 hrs pre-incubation with RBN2397?) could be included in Fig. 2C - it distinguishes the requirement of PARP7 enzyme activity vs. PARP7 protein level for spheroids formation. Also, long term RBN2397 treatment (48 hrs) mimics the PARP7 KO condition in ISGF3 signaling (Supplementary Fig. 5), this could be expanded to other experiments where there are different outcomes from PARP7 KO vs WT with RBN2397 (24 hrs) treatment.
Author Response
Reviewer 3
Since the re-introduction of PARP7 to cells usually takes a longer time, be it lentiviral (authors indicated that their department does not allow this practice there) or gene knocking in, Rasmussen et al have satisfied most of my concerns for which the experiments could be done in a relatively short time. However, it seems that RBN2397 condition (maybe needs 48 hrs pre-incubation with RBN2397?) could be included in Fig. 2C - it distinguishes the requirement of PARP7 enzyme activity vs. PARP7 protein level for spheroids formation. Also, long term RBN2397 treatment (48 hrs) mimics the PARP7 KO condition in ISGF3 signaling (Supplementary Fig. 5), this could be expanded to other experiments where there are different outcomes from PARP7 KO vs WT with RBN2397 (24 hrs) treatment.
Response: We thank the reviewer for his/her positive feedback. The spheroid data provided in Fig 2C show differences between untreated WT and the two different P7ko clones. We cannot exclude that 48 h or longer treatment with RBN2397 might also reduce spheroid formation similar to that observed for Parp7KO cells. However, we will be unable to do those in a timely manner because of a couple of external factors. (1) It is common holiday time in Norway and all of my lab (including myself) will be on vacation for at least 3 weeks and (2) the University of Oslo is upgrading the electrical box that runs the entire building that we work in. This means that the power will be cut to all nonessential instruments for a least 2 weeks (week 29 and 30). There could of course be problems once everything is tuned back on, so there is not guarantee that we will be able to start work from day 1.
In our previous response to other reviewers’ comments after round 1 we mentioned that we have started a study to compare that ability of the 3 known Parp7 selective inhibitors, RBN2397, KMR and I-1, across a panel PARP7KO cells and MEFs isolated from Parp7H532A mice. We feel it is important to compare multiple selective PARP7 inhibitors under different stimulation conditions and in the presence and absence of PARP7 protein. This is such a critical topic that we feel it deserves its own manuscript rather than be included in the current study. We are not sure if this reviewer had access to our response to other reviewers’ comments, so we included this again here.
To address some of the differences between WT and Parp7KO cells we provided western blot data after 48 h treatment with RBN (Supplementary Figure S6 - previously S5) as noted by the reviewer. To expand this to all experiments by repeating treatments for 24h and/or 48h would not be feasible for the same reasons that are listed above. However, we have determined Ifnb1 and Cxcl10 levels in WT and Parp7KO EO771 cells as well as WT and Parp7H532A MEFs treated with DMSO or RBN2397 for 24h. Treatment with RBN-2397 for 24h resulted in similar expression levels of Ifnb1 compared with those observed in untreated Parp7KO cells, but higher levels of Cxcl10. This was similar to that observed with DMXAA+RBN treatment in Fig 2F. Comparable findings were also observed in 24h treated WT and Parp7H532A MEFs. These new data have been added in Supplementary Figure S4A-D. New text summarizing the findings has been added to page 8 395-398 and page 9 429 and 430. All other supplementary figures have been re-numbered to reflect the addition of new supplementary figure S4. The manuscript text has also been updated to reflect these changes.
Reviewer 4 Report
The authors have addressed all the relevant points raised, and therefore I believe that the manuscript can be considered for publication in Cancers
Author Response
Reviewer 4
The authors have addressed all the relevant points raised, and therefore I believe that the manuscript can be considered for publication in Cancers
Response: We thank the reviewer for his/her constructive comments and positive feedback about our manuscript.
Round 3
Reviewer 3 Report
I wish the authors could complete the experiments I suggested, but I do understand their situation. Overall, this is a good paper in my opinion.